# Evaluation of the Efficiency in Public Health Centers in Greece Regarding the Human Resources Occupied: A Bootstrap Data Envelopment Analysis Application

**DOI:** 10.3390/ijerph19031597

**Published:** 2022-01-30

**Authors:** Anastasios Trakakis, Miltiadis Nektarios, Styliani Tziaferi, Panagiotis Prezerakos

**Affiliations:** 1Faculty of Health Sciences, Department of Nursing, University of Peloponnese, 22100 Tripoli, Greece; sttziaf@hotmail.com (S.T.); prezerpot@gmail.com (P.P.); 2Faculty of Finance and Statistics, Department of Statistics and Insurance Science, University of Piraeus, 18534 Athens, Greece; nektar@unipi.gr

**Keywords:** health center’s efficiency, human resources, data envelopment analysis, bootstrap, internal validity, primary health care, C14, C32, C52, C67, I10, I18, J08

## Abstract

In this paper, the overall efficiency of health centers in Greece is measured by applying the input-oriented model of Data Envelopment Analysis. In addition, four different models were subjected to the input-oriented Data Envelopment Analysis to investigate the contribution of each category of human resources to the efficiency results of the health centers. The bootstrap technique was performed to generate confidence intervals for the models. Data for 155 health centers in Greece were provided by the Ministry of Health. The health centers submitted in the analysis obtained an average efficiency value of 0.932. The average results of the partial models in terms of each input show that the efficiency values achieved by the health centers are mostly influenced by the number of physicians and the number of managers employed. The second factor influencing the efficiency values of the health centers are the number of nursing staff occupied in the health centers. Non-medical staff employed in the health centers had the least contribution to the efficiencies measured. This paper provides important information for the stakeholders and the Government of Greece so as to better allocate the personnel employed in primary health care according to the efficiencies attained by the health centers.

## 1. Introduction

In the last decade, the World Health Organization (WHO) has focused on a strategy to strengthen and integrate primary health care, considering its major contribution to the national health system of every country. OECD countries, including Greece, have committed to follow the strategy set by WHO [1].

Moreover, as was stated by Myloneros et al. [2], Greece faces high overall health care expenditures and high rates of chronic diseases, which can be addressed by a strong and integrated primary health care system. In addition, as a result of the economic crisis and funding cuts to the National Health System in Greece during 2010–2018, health expenditures in 2017 were 1.8% lower than the average of OECD countries [3,4], highlighting the already existing inadequate primary health care system [5], which is struggling with problems in the distribution of health services, productivity, and administration [6,7,8].

A reform was introduced to strengthen the National Health System by creating an integrated primary health care system, which included the fragmentation of primary health care services and the introduction of a two-pillar primary health care system from 2017 to 2020 [9,10]. The first pillar included the creation of Local Health units (TOMY) and the concept of the “family physician”, who corresponds to a group of people and guides them through the health system, while the second pillar included all health centers in Greece. The reform was established by Law 4486/2017 and predicted the creation of 239 TOMY, but only 127 TOMY eventually opened. Moreover, health centers upgraded their role to the primary health care system, including more complex medical interventions and the supervision of TOMY. The 127 TOMY did not function as it was predicted, since they opened in the mainland of Greece and rural areas were hardly covered [2,9,10].

People showed no confidence in the reform and were dissatisfied with the primary health care system. The reform failed in the first 3 years due to lack of planning and organization, although it was based on an attempt to create an integrated model of primary health care [2,9,10].

The failure of the reform led the whole primary health care service to encumber the health centers in Greece and the personnel occupied in them. Considering that there are 207 public health centers in operation throughout Greece and that primary health care is mostly delivered by them, the evaluation of their efficiency is critical, providing information to the stakeholders and the Government.

This paper focuses on the impact of each type of human resource on the efficiencies of the 155 health centers. This is because human resources are a highly specialized factor in the health sector and two-thirds of financial resources are spent on labor. Moreover, human resources are the main factor that ensures quality in health care services [11]. The distribution of human resources according to the needs of health centers is crucial for achieving a high level of efficiency. In addition, in the period 2016–2018, primary health care in Greece faced the impact of the economic crisis and the outcomes of the reform. This analysis estimates the efficiency of health centers during this unstable period, which was also the period just before the outbreak of the COVID-19 pandemic.

The study focuses on investigating which of the human resources used as inputs, regarding the outputs produced, mostly affects the efficiency of health centers [11]. In addition, the analysis included 155 of the 207 health centers that operated in Greece during 2016–2018, due to missing data for the other 52 health centers. Ninety percent of the health centers included in the analysis operated in rural areas [10].

## 2. Materials and Methods

### 2.1. Literature Review

Efficiency can be measured by parametric and non-parametric approaches. Data Envelopment Analysis (DEA) is a non-parametric approach, which is widely used in health care studies because it avoids the difficulty of assuming and predefining a functional form [12]. The nonparametric approach DEA is mainly applied to estimate efficiencies in health care studies [13,14]. One of the benefits of the DEA method is the ability to use multiple inputs and outputs, even if they have different measures [13].

Farrell, in 1957 [15], introduced modern efficiency following the work of Debreu and Koopmans in 1951 [16,17]. Farrell measured the efficiency of one firm using multiple inputs and analyzed and divided the measured efficiency into technical and allocative efficiency.

In 1978, following Farrell’s definition of efficiency, Charnes, Cooper, and Rhodes presented the linear programming DEA method, in an attempt to construct a nonparametric frontier. The firms in DEA are called decision-making units (DMUs) and use inputs to produce outputs. The inputs used and the outputs produced are used as data to measure the efficiency of the DMUs. The efficient DMUs are on the constructed frontier while the inefficient DMUs are below the constructed frontier [18,19]. DEA maximizes the efficiency of DMUs according to the best practice of the process of transforming inputs into outputs [5].

Two linear programming problems can be solved when measuring efficiency using DEA. The first problem is to minimize the inputs used while keeping the outputs produced constant when computing efficiencies, while the other problem is to maximize the outputs produced while keeping the inputs used constant. The first is the input orientation of DEA, while the other is the output orientation of DEA [12,18]. The orientations define the linear programming equation to construct the frontier and measure the technical efficiencies of the health centers.

Moreover, the two assumptions of DEA—Constant Returns to Scale (CRS), introduced by Charnes, Cooper, and Rhodes in 1978, and Variable Returns to Scale (VRS), introduced by Banker, Charnes, and Cooper in 1984—lead to different results in terms of efficiency. The first method (CRS) assumes that all DMUs operate at an optimal level of efficiency, while the second method (VRS) assumes that there are scale inefficiencies and that not all DMUs operate at an optimal level, so imperfect competition is accounted for in the analysis [20,21].

Most studies on healthcare in Greece, such as Fragkiadakis et al. 2016 [22], Androutsou et al. 2011 [5], Xenos et al. 2017 [23], Dimas et al. 2012 [14], and Prezerakos et al. 2007 [24] focused on estimating the efficiency and productivity of hospitals in Greece. Although these studies measure the efficiency and productivity of hospitals in Greece, there are no studies for the primary health care system, which has received more attention recently. In a previous paper for primary health care in Greece, the efficiency of 198 health centers was estimated for 2018 [21]. In contrast, this study estimates the efficiency of health centers considering the mean values of inputs and outputs for 2016–2018 and also examines the impact of human resources on the efficiencies attained. Human resources are the most important factor providing health care services in the health centers and also affect the quality of the health services. Thus, it is of utmost interest to investigate which of the human resources mainly affect the efficiencies of the health centers.

The input-oriented DEA is used to evaluate the efficiency of health centers because the outputs cannot be precisely defined in advance. In contrast, inputs can be more easily defined and controlled [25,26,27]. Moreover, efficiency is calculated over a multi-year period to ensure greater stability of the data used (mean values) and avoid outlined observations. In addition, the results of the health centers’ efficiencies will be more realistic. The efficiencies calculated under the DEA method were also subjected to a bootstrap analysis so as to generalize the model and create confidence intervals for 1000 samples, which allows a more accurate estimation on the efficiencies.

### 2.2. Model Specification—Data Envelopment Analysis

The target of the analysis is to estimate the efficiencies of 155 health centers in Greece, in 2016–2018 [10], to construct confidence intervals about the measured efficiencies, and to investigate the contribution of each human resource to the efficiency of each health center. The mathematical concept of the input-oriented DEA under CRS and VRS is summarized below.

Under the hypothesis that there are N DMUs subjected to the analysis, using K inputs to produce M outputs, two matrices K*N and M*N are created. The K*N matrix represents the inputs used and is referred to as X, while the M*N matrix represents the outputs produced and is referred to as Y [28]. To measure the efficiency of each DMU, T.J. Coelli (1996) presented a mathematical linear programming equation calculating the ratio of all outputs over all inputs:“min_θ,λ_ θ,
s.t.
−y_i_ + Yλ ≥ 0,
θx_i_ − Xλ ≥ 0,
λ ≥ 0.

The symbol θ is a scalar, and λ is an N*1 vector of constants” [28].

In the DEA method, “the symbol θ takes values within the closed interval [0, 1] and represents the efficiencies of the DMUs. The problem to be solved is to estimate the values of θ. DMUs, with values of θ equal to 1, are operating at an optimal efficiency level, while DMUs, with values of θ less than 1, are inefficient. The linear programming problem has to be solved N times for each DMU” [10,28].

The linear programming problem above is under the assumption of CRS. In contrast, the linear programming problem under the VRS assumption has one more constraint, N1′λ = 1, added to the CRS linear programming problem. By adding this constraint to the model CRS, “scale efficiency effects are calculated, and the technical efficiency is divided into pure technical efficiency and scale efficiency for each DMU. N1 represents an N*1 vector of ones” [10,21,28].

Data for three years (2016–2018) are used to determine the mean values for the 3 years of the study. The DEA analysis is performed by CRS and VRS assumptions to measure the efficiencies of the DMUs, including all inputs used to produce all outputs, which represent all the personnel occupied to deliver all the medical actions of health centers. In addition, four other models were constructed, including only one input used to produce all outputs, so as to examine the contribution of each input to the overall efficiency of each health center.

In addition, in order to calculate the standard deviation and confidence intervals of health centers’ efficiencies, the bootstrap method was applied to the results of DEA. According to the literature, the bootstrap method is a computer technique that aims to generate data from a sample. The sample used is the efficiencies of the 155 health centers, and the bootstrap method used generates 1000 samples that have the same size as the original sample and are based on the values of the first sample, which allows the creation of confidence intervals [11,29].

### 2.3. Data

The values of the data used as inputs and outputs in this study represent the average values of inputs and outputs in 2016–2018. The average of three years was intentionally chosen to avoid including misleading information about the staff employed in the health centers and the procedures delivered by them in the analysis, capturing the most up-to-date inputs used and outputs produced by the health centers before the outbreak of the COVID-19 pandemic in 2019 [11].

Random estimates and the likelihood of bias were avoided by excluding from the analysis the 52 health centers with missing data. Thus, 155 were included in the analysis, and their efficiencies were calculated during the period 2016–2018 [10].

The health centers included in the analysis represent 74.87% of the health centers operating in Greece and are distributed throughout the country, ensuring both homogeneity and discriminatory power between efficient and inefficient units [30,31]. Moreover, ninety percent of the health centers in the analysis operated in rural areas [10].

The requirements to perform DEA are satisfied, ensuring comparability and validation in measuring the health centers’ efficiencies. Firstly, “the health centers submitted into the analysis use the same categories of inputs and produce the same categories of outputs, contrasting only in the quantities used. Secondly, at least one DMU in the sample consumes and produces each input and output, and each DMU in the sample consumes at least one input and produces at least one output” [10,32,33,34,35].

In the DEA analysis, four inputs and 12 outputs were used in order to measure the efficiencies of the health centers. The four inputs refer to the total staff occupied by the health centers, while the 12 outputs refer to the total health services provided by them [10]. In Table 1, the inputs and outputs used are presented.

The data of the inputs and the outputs, submitted for the DEA analysis in the process of calculating the efficiencies of the 155 health centers in Greece, are the averages of the 2016–2018 period. The descriptive statistics in Table 2 show the minimum, maximum, mean, and standard deviation for each input and output.

## 3. Results

### 3.1. Efficiency of the Total Model (M0), Which Includes all Inputs and Outputs in the DEA Analysis under Both the CRS and VRS Assumptions

The input-oriented DEA, under the CRS and VRS assumption, was performed by the DEAP ver2.1 program to measure the efficiency of each of the 155 health centers. The bootstrap technique was performed to generate confidence intervals for the models [28].

The summary Table 3 of the total model (M0) shows the results of the mean technical efficiency of the 155 health centers, under the CRS and VRS assumptions.

In Table 4 the correlation between the efficiencies, calculated under CRS and VRS assumptions, was estimated by a Spearman-rank correlation test [36].

The statistically significant Spearman-rank correlation coefficient is 0.646, showing a high degree of correlation between the results of the two methods.

### 3.2. Different Models—To Study the Contribution of Human Resources to the Overall Efficiency of Health Centers

To study the contribution of human resources to the efficiency of health centers depending on the specialization of the staff, this paper creates four other models. Each model includes all outputs produced by health centers, but only one input at a time. Considering the created models, the efficiencies were calculated under the CRS and VRS assumptions of input-oriented DEA. Consequently, partial efficiencies are estimated from the perspective of each input (Input1, Input2, Input3, and Input4) compared to all outputs, under both the CRS and the VRS assumption. All these partial efficiencies aim to better explain the contribution of each category of human resource to the efficiencies of the health centers [11].

Table 5 shows the different models created to estimate the efficiencies of the health centers using DEA, in order to study the effect of each input in the production of all outputs. The symbol “X” under the inputs and outputs in the table indicates that the input or output is present in the model accordingly. Table 5 is used to illustrate the models and the various inputs used.

M0 is the model that acknowledges all inputs and outputs in estimating health centers’ efficiencies, while the other models use only one input to produce all outputs, in an attempt to measure the effect of each input on the efficiencies. Thus, M1 uses as input the number of managers, M2 the number of physicians, M3 the number of nursing staff, and M4 the number of non-medical staff employed in the health centers.

The average efficiency values and confidence intervals with bootstrap technique in the efficiencies under each of the models are presented in Table 6.

As was expected, the efficiencies calculated under the CRS assumption were lower than those calculated under the VRS assumption.

The M1_VRS and M2_VRS models achieve the highest efficiency scores compared to the efficiency of the total model M0 (0.14 difference), suggesting that managers and physicians contribute the most to the overall efficiency of health centers. The Appendix A summarizes the efficiency scores of the 155 health centers for all models and all assumptions. Indices with values of one represent technically efficient health centers, while values below one indicate inefficiencies [37]. In addition, the inefficient health centers with efficiency scores below the mean efficiency scores should consider reducing their inputs or producing more outputs. Table 6 presents the 95% confidence intervals (a = 5%) of the models according to the bootstrap results on the calculated efficiencies, indicating the upper and lower bounds of the average values of the efficiencies. I. Vrabková and I. Vaňková in 2021 used the confidence interval to specify the inefficiency rate (IR), as follows:Mild inefficiency: 1 > R ≥ Upper;Moderate inefficiency: Upper > IR ≥ Lower;Strong inefficiency: IR < Lower [11].

Table 7 and Figure 1 below, show the number of technically efficient, mild inefficient, moderate inefficient, and strong inefficient health centers for each model.

It is shown that the most efficient health centers are obtained under the VRS assumption of the total model (M0). For the partial models (M1, M2, M3, and M4), the most efficient health centers are obtained for the model that has as input the number of managers occupied in the health centers (64 health centers—VRS assumption model), while the least in number efficient health centers (10 health centers) are obtained for the model that has as input the number of non-medical staff occupied in the health centers (CRS assumption model). Additionally, in the model that takes the non-medical staff as input, the most “strong inefficient” health centers are obtained (84 health centers).

The average results and the inefficiency rate of the partial models, with respect to each input (Input1—number of managers, Input2—number of physicians, Input3—number of nursing staff, and Input4—number of non-medical staff) show that the efficiency levels achieved by the health centers are most influenced by the number of employed managers, followed by the number of physicians and the number of nursing staff. The number of non-medical staff employed by health centers in Greece has the least influence on the efficiency levels. However, it is important to highlight the fact that the model using as input the number of employed managers indicates both many efficient health centers (R = 1) and many inefficient health centers (R < Lower), while there are very few health centers with mild or moderate efficiency. In contrast, the model that takes into account the number of physicians employed in health centers attain fewer but still many efficient health centers (R = 1) and also many health centers with mild and moderate efficiency (1 > R ≥ Upper, Upper > R ≥ Lower). In models M1 and M2, the sum of the number of efficient and mild inefficient health centers is approximately equal (two health centers difference).

In order to examine the correlation between the different models under both the VRS and CRS assumptions, the Spearman rank correlation coefficients were calculated and presented in Table 8.

The statistically significant coefficients show the correlation between the different models under the assumptions of CRS and VRS and also ensure the internal validity of the model, which will be further investigated.

### 3.3. Model Validation—Internal Validity

Internal validity compares the effects on health center efficiency when different inputs and outputs are used. In DEA, efficiencies measured by different models cannot be compared directly, but comparison of efficiencies can be applied using Spearman rank correlation tests [10,38,39,40,41]. In this paper, four different models under the assumption of VRS and CRS have already been used to study the effects of each input on the efficiency of health centers. Two more models were created under the VRS assumption and submitted to DEA analysis. The models are shown in Table 9 and are based on the VRS assumption. To illustrate the models used, the symbol “X” is used to indicate the presence of inputs and outputs for each model. 

The first five models in the table (M0, M1, M2, M3, and M4) are the models that had already been used for this study, while the other two models (M5 and M6) are the new models created. Model M5 excludes the outputs, chronic disease cases and emergencies faced, while model M6 excludes the inputs non-medical staff and managers, and the outputs dental procedures, transcriptions, and Mantoux test. The new M5 and M6 models, which exclude both outputs and inputs from the analysis, were created to further support the test of internal validity, whereas the M1, M2, M3, and M4 models excluded only inputs. In this way, Spearman rank correlation coefficients are additionally calculated for two more models, providing a stronger and more comprehensive test of internal validity. The outputs and inputs excluded in models M5 and M6 were randomly selected so as not to bias the results.

The efficiencies calculated for the six different models, under input-oriented VRS DEA, were submitted into a Spearman’s-rank correlation test, in order to check the validity of the model. The Spearman-rank correlation coefficients are presented in Table 10.

Spearman’s rank correlation coefficients are statistically significant, indicating internal validity for the model under the VRS assumption.

## 4. Discussion

Data of inputs and outputs for the period 2016–2018 were provided by the Ministry of Health. Attempt to collect data for all health centers operating in Greece was made, but eventually there was availability of data for 155 of them, due to the change of the system the Ministry of Health collects data in 2016 [10].

The DEA method was used to estimate the efficiency of health centers in Greece during 2016–2018. Moreover, after estimating the efficiency, four other models were subjected to DEA analysis to investigate the influence of human resources on the calculated efficiencies, since health centers are labor-intensive units and health services are delivered by the personnel occupied in them. Meaningful results were obtained on the overall efficiency of health centers in terms of human resources used. Moreover, the evaluation of efficient health centers is crucial as it points out the viability of each health center [10,14]. This study, considering the mean values of the data used as inputs and outputs to calculate the efficiencies, over the period 2016–2018, found an average efficiency value of 0.932.

The contribution of each input to health center efficiency was examined by creating four additional models with one input each while all outputs were produced. The model using the number of managers as an input and the model using the number of physicians employed at the health centers as an input, both under the VRS assumption, achieved the highest mean efficiency scores, approximately 0.68. The model using non-medical staff as an input achieved the lowest mean efficiency score. As expected, the efficiency scores calculated under the CRS assumption were lower than the efficiency scores calculated under the VRS assumption. Considering the results of the inefficiency rates calculated for each of the models in this study under the VRS assumption, the analysis shows that the model using the number of employed managers as input has the most efficient health centers (64 health centers), followed by the model using as input the number of employed physicians (39 efficient health centers). The least in number efficient health centers resulted from the model that considered the number of employed non-medical staff as an input (31 health centers). The most “mild inefficient” health centers resulted from the model that considered the number of physicians as an input to the production of all outputs (31 health centers). The results of the inefficiency rates are consistent with the results of the DEA applications. According to this study, when trying to optimize human resources in health centers, it appears coherent to extend the number of physicians and managers whereas lessening the number of non-medical staff, taking under consideration the necessities and needs of each health center. Moreover, it is known that the number of physicians is positively related to the quality of health services.

Health centers that are operating inefficiently ought to follow the best practices of the efficient ones [10,11,21]. The inefficient health centers should individually consider hiring more physicians and managers and reducing non-medical staff to follow the example of the efficient health centers. However, in restructuring human resources, health centers should consider the impact that it would have on their operation and the quality of health services provided.

## 5. Conclusions

This paper may contribute to improve the overall efficiency of health centers in Greece. Furthermore, valuable information can be delivered for the National Health Care System to harmonize the available human resources according to the necessities of each health center, as well as for managers and stakeholders in primary health care to ensure the essential personnel for the optimum operation of the health centers [10].

It is recognized that health systems with solid people-centered, integrated, and organized primary health care are more effective [2,10]. To progress efficiencies, it is suggested that the stakeholders in Greece’s National Health System better allocate the staff employed in health centers and apply continuous development and education for the professional staff. Considering the results in the economic crisis and the funding cuts in the Greek health system, as well as the reform that has been carried out, it is important to distribute the staff employed in the health centers in a way that maximizes their efficiency. If we achieve better results, people will embrace primary health care in Greece, relieving hospitals from facing chronic disease cases and simple medical procedures that can be managed through primary health care system. By optimally distributing staff in health centers and improving their efficiency, the entire health system will be strengthened.

An area for further research could be the study of other exogenous factors that may influence the efficiency of health centers through Tobit regression analysis, as well as an evaluation of the efficiency of health centers considering economic measurements. This was outside the scope of this study and was therefore not considered, although it is important for the viability and efficiency of health centers. In addition, a study evaluating the efficiency of TOMYs and the contribution of each category of human resources to that efficiency would provide meaningful results. These results, together with the findings of this study, could be used to reform and advance the entire primary health care system in Greece in terms of the allocation of human resources used.

The program DEAP version 2.1 for Windows by T. Coelli (1996) was used [28] to estimate the efficiencies of the health centers. Statistics were performed by the IBM SPSS program.

## Figures and Tables

**Figure 1 ijerph-19-01597-f001:**
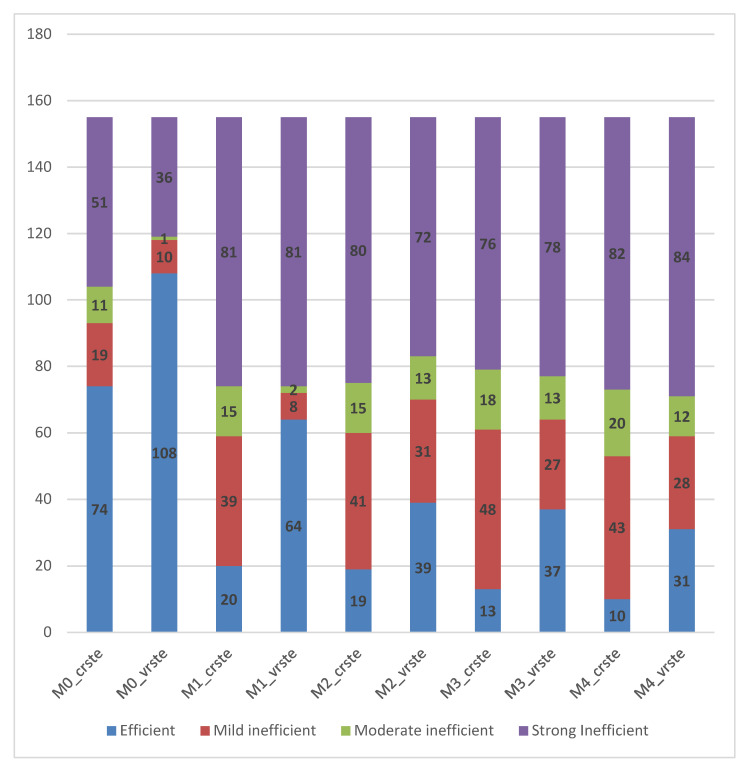
Efficient, mild inefficient, moderate inefficient, and strong inefficient health centers.

**Table 1 ijerph-19-01597-t001:** Data used for the DEA analysis.

N/N	Data	Description
Output_1	Nursing operations	Average number applied (2016–2018)
Output_2	Microsurgeries	Average number applied (2016–2018)
Output_3	Dental procedures	Average number applied (2016–2018)
Output_4	Chronic disease cases	Average number faced (2016–2018)
Output_5	Emergencies	Average number faced (2016–2018)
Output_6	Regular incidents	Average number faced (2016–2018)
Output_7	Urgent incidents	Average number faced (2016–2018)
Output_8	Transcriptions	Average number given (2016–2018)
Output_9	Biopathological and laboratory exams	Average number applied (2016–2018)
Output_10	Mantoux test	Average number applied (2016–2018)
Output_11	Vaccinations for adults	Average number applied (2016–2018)
Output_12	Vaccinations for kids and teenagers	Average number applied (2016–2018)
Input_1	Number of managers	Average number employed (2016–2018)
Input_2	Number of physicians	Average number employed (2016–2018)
Input_3	Number of nursing staff	Average number employed (2016–2018)
Input_4	Number of non-medical staff	Average number employed (2016–2018)

**Table 2 ijerph-19-01597-t002:** Descriptive statistics.

Data (Inputs-Outputs)	Minimum	Maximum	Mean	Std. Deviation
Nursing operations	0	31,376	4177.67	4441.781
Microsurgeries	0	17,992	475.19	1526.713
Dental procedures	0	11,417	1191.77	1630.295
Chronic disease cases	0	34,036	4553.23	7154.341
Emergency cases	0	20,443	1664.09	2124.164
Regular incidents	0	56,402	13,645.38	9716.818
Urgent incidents	920	38,239	9129.65	7394.567
Transcriptions	0	61,460	11,679.42	8786.179
Bio-pathological and Laboratory exams	0	146,277	15,463.58	21,816.909
Test Mantoux	0	855	64.82	107.757
Vaccinations applied for adults	0	4155	498.87	605.530
Vaccinations applied for Kids and teenagers	0	2656	519.34	586.265
Number of managers	1	11	2.57	1.673
Number of physicians	2	32	9.46	6.360
Number of nursing staff	2	51	16.25	9.820
Number of non-medical staff	1	32	7.38	4.285

**Table 3 ijerph-19-01597-t003:** Summary table of the efficiencies (total model).

Statistics	CRS	VRS
Arithmetic mean	0.82046	0.93188
Geometric mean	0.82	0.932
Std. Deviation	0.234338	0.136112
Minimum	0.243	0.260
Maximum	1.000	1.000

**Table 4 ijerph-19-01597-t004:** Spearman-rank correlation test.

Model Type	CRS Total Model	VRS Total Model
CRS total model	1.000	0.646 *
VRS total model	0.646 *	1.000

* Corr. Sign. 0.01 level (2-tailed). Ν:155.

**Table 5 ijerph-19-01597-t005:** Different models.

Models/Variables	O1	O2	O3	O4	O5	O6	O7	O8	O9	O10	O11	O12	I1	I2	I3	I4
M0_CRS/VRS	Χ	Χ	Χ	Χ	Χ	Χ	Χ	Χ	Χ	Χ	Χ	Χ	Χ	Χ	Χ	Χ
M1_CRS/VRS	Χ	Χ	Χ	Χ	Χ	Χ	Χ	Χ	Χ	Χ	Χ	Χ		Χ	Χ	Χ
M2_CRS/VRS	Χ	Χ	Χ	Χ	Χ	Χ	Χ	Χ	Χ	Χ	Χ	Χ	Χ		Χ	Χ
M3_CRS/VRS	Χ	Χ	Χ	Χ	Χ	Χ	Χ	Χ	Χ	Χ	Χ	Χ	Χ	Χ		Χ
M4_CRS/VRS	Χ	Χ	Χ	Χ	Χ	Χ	Χ	Χ	Χ	Χ	Χ	Χ	Χ	Χ	Χ	

Note: O1: Nursing operations, O2: Microsurgeries, O3: Dental procedures, O4: Chronic disease cases, O5: Emergencies, O6: Regular incidents, O7: Urgent incidents, O8: Transcriptions, O9: Biopathological and laboratory exams, O10: Mantoux test, O11: Vaccinations for adults, O12: Vaccinations for kids and teenagers, I1: Number of managers, I2: Number of physicians, I3: Number of nursing Staff, I4: Number of non-medical staff.

**Table 6 ijerph-19-01597-t006:** Descriptive Statistics of the efficiencies calculated under the eight DEA analyses performed.

Models		Stat.	Bootstrap *			
			Bias	Std. Error	95% Confidence Interval	
					Lower	Upper
MO_CRS	Mean	0.82046	−0.00027	0.01892	0.78254	0.85619
	Std. Deviation	0.234338	−0.000838	0.012096	0.208610	0.254964
MO_VRS	Mean	0.93188	−0.00034	0.01112	0.90906	0.95162
	Std. Deviation	0.136112	−0.000488	0.014381	0.108685	0.164017
M1_CRS	Mean	0.49852	−0.00059	0.02198	0.45914	0.54502
	Std. Deviation	0.275431	−0.001470	0.011573	0.250244	0.296290
M1_VRS	Mean	0.68973	−0.00023	0.02343	0.64653	0.73417
	Std. Deviation	0.291907	−0.000951	0.007704	0.275718	0.305765
M2_CRS	Mean	0.57935	0.00051	0.02048	0.53783	0.62132
	Std. Deviation	0.260694	−0.000680	0.009663	0.239598	0.277406
M2_VRS	Mean	0.68061	0.00053	0.01987	0.64151	0.72187
	Std. Deviation	0.253099	−0.000768	0.008542	0.235300	0.269455
M3_CRS	Mean	0.49004	−0.00007	0.02173	0.44610	0.53466
	Std. Deviation	0.270005	−0.001039	0.012431	0.244536	0.294106
M3_VRS	Mean	0.57066	−0.00001	0.02514	0.52114	0.61963
	Std. Deviation	0.310264	−0.001232	0.009881	0.290893	0.327554
M4_CRS	Mean	0.42086	0.00000	0.02240	0.37935	0.46659
	Std. Deviation	0.278288	−0.000682	0.013285	0.250353	0.302795
M4_VRS	Mean	0.51788	−0.00002	0.02520	0.46881	0.56558
	Std. Deviation	0.314267	−0.001064	0.010558	0.292209	0.332964

* Bootstrap specifications. sampling method: simple, number of samples: 1000, confidence interval level: 95.0%.

**Table 7 ijerph-19-01597-t007:** Efficient, mild inefficient, moderate inefficient, and strong inefficient health centers.

Models	Efficiency	Mild	Moderate	Strong
M0_CRS	74	19	11	51
M0_VRS	108	10	1	36
M1_CRS	20	39	15	81
M1_VRS	64	8	2	81
M2_CRS	19	41	15	80
M2_VRS	39	31	13	72
M3_CRS	13	48	18	76
M3_VRS	37	27	13	78
M4_CRS	10	43	20	82
M4_VRS	31	28	12	84

**Table 8 ijerph-19-01597-t008:** Correlation coefficient.

Spearman’s Rho	M0_CRS	MO_VRS	M1_CRS	M1_VRS	M2_CRS	M2_VRS	M3_CRS	M3_VRS	M4_CRS	M4_VRS
M0_CRS	1	0.646 **	0.624 **	0.214 **	0.645 **	0.612 **	0.701 **	0.711 **	0.740 **	0.686 **
M0_VRS	0.646 **	1	0.476 **	0.570 **	0.389 **	0.486 **	0.441 **	0.487 **	0.412 **	0.451 **
M1_CRS	0.624 **	0.476 **	1	0.719 **	0.486 **	0.474 **	0.489 **	0.520 **	0.454 **	0.432 **
M1_VRS	0.214 **	0.570 **	0.719 **	1	0.132	0.289 **	0.223 **	0.304 **	0.085	0.193 *
M2_CRS	0.645 **	0.389 **	0.486 **	0.132	1	0.866 **	0.364 **	0.361 **	0.410 **	0.362 **
M2_VRS	0.612 **	0.486 **	0.474 **	0.289 **	0.866 **	1	0.370 **	0.446 **	0.412 **	0.457 **
M3_CRS	0.701 **	0.441 **	0.489 **	0.223 **	0.364 **	0.370 **	1	0.935 **	0.447 **	0.461 **
M3_VRS	0.711 **	0.487 **	0.520 **	0.304 **	0.361 **	0.446 **	0.935 **	1	0.494 **	0.557 **
M4_CRS	0.740 **	0.412 **	0.454 **	0.085	0.410 **	0.412 **	0.447 **	0.494 **	1	0.913 **
M4_VRS	0.686 **	0.451 **	0.432 **	0.193 *	0.362 **	0.457 **	0.461 **	0.557 **	0.913 **	1

* Corr. Sig. at 0.05 level (2-tailed). ** Corr. Sig. at 0.01 level (2-tailed).

**Table 9 ijerph-19-01597-t009:** Different models for internal validity check.

Models/Variables	O1	O2	O3	O4	O5	O6	O7	O8	O9	O10	O11	O12	I1	I2	I3	I4
M0_VRS	Χ	Χ	Χ	Χ	Χ	Χ	Χ	Χ	Χ	Χ	Χ	Χ	Χ	Χ	Χ	Χ
M1_VRS	Χ	Χ	Χ	Χ	Χ	Χ	Χ	Χ	Χ	Χ	Χ	Χ		Χ	Χ	Χ
M2_VRS	Χ	Χ	Χ	Χ	Χ	Χ	Χ	Χ	Χ	Χ	Χ	Χ	Χ		Χ	Χ
M3_VRS	Χ	Χ	Χ	Χ	Χ	Χ	Χ	Χ	Χ	Χ	Χ	Χ	Χ	Χ		Χ
M4_VRS	Χ	Χ	Χ	Χ	Χ	Χ	Χ	Χ	Χ	Χ	Χ	Χ	Χ	Χ	Χ	
M5_VRS	Χ	Χ	Χ			Χ	Χ	Χ	Χ	Χ	Χ	Χ	Χ	Χ	Χ	Χ
M6_VRS	Χ	Χ		Χ	Χ	Χ	Χ		Χ		Χ	Χ		Χ	Χ	

Note: O1: Nursing operations, O2: Microsurgeries, O3: Dental procedures, O4: Chronic disease cases, O5: Emergencies, O6: Regular incidents, O7: Urgent incidents, O8: Transcriptions, O9: Biopathological and laboratory exams, O10: Mantoux test, O11: Vaccinations for adults, O12: Vaccinations for kids and teenagers, I1: Number of managers, I2: Number of physicians, I3: Number of nursing staff, I4: Number of non-medical staff.

**Table 10 ijerph-19-01597-t010:** Internal Validity check—Spearman-rank correlation coefficients.

Spearman’s Rho	M0_VRS	M1_VRS	M2_VRS	M3_VRS	M4_VRS	M5_VRS	M6_VRS
M0_VRS	1	0.570 **	0.486 **	0.487 **	0.451 **	0.906 **	0.524 **
M1_VRS	0.570 **	1	0.289 **	0.304 **	0.193 *	0.606 **	0.202 *
M2_VRS	0.486 **	0.289 **	1	0.446 **	0.457 **	0.460 **	0.672 **
M3_VRS	0.487 **	0.304 **	0.446 **	1	0.557 **	0.434 **	0.661 **
M4_VRS	0.451 **	0.193 *	0.457 **	0.557 **	1	0.386 **	0.370 **
M5_VRS	0.906 **	0.606 **	0.460 **	0.434 **	0.386 **	1	0.477 **
M6_VRS	0.524 **	0.202 *	0.672 **	0.661 **	0.370 **	0.477 **	1

* Corr. sign. at 0.05 level (2-tailed). ** Corr. Sign. at 0.01 level (2-tailed).

## Data Availability

The data that support the findings of this study are available from Bi-health (Ministry of Health), but restrictions apply to the availability of these data, which were used under license for the current study and so are not publicly available. Data are, however, available from the authors upon reasonable request and with permission of the Ministry of Health.

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
