# Peer review of "Evaluation of the Efficiency in Public Health Centers in Greece Regarding the Human Resources Occupied: A Bootstrap Data Envelopment Analysis Application"

_ijerph, 2022, doi:10.3390/ijerph19031597_

Round 1
Reviewer 1 Report
Dear Authors,
Although the interest of the subject, this manuscript needs some improvements, namely:
- Discussion section needs to be improved.
Authors should develop a deeper confrontation between the obtained results and the literature revision implemented above. - In the conclusion section authors should highlight the main theoretical and practical contributions of their study to this research field.
Author Response
Dear reviewer,
The suggested improvements have been made into the manuscript. In the Discussion section, the results have been further analyzed and presented and compared with the literature used in the paper. Also, in the conclusion, the theoretical and practical contribution of this study has been presented and suggestions for further research have been made. Thank you for your time and critical interventions, which have been necessary for our work.
Best Regards,
Anastasios Trakakis
Reviewer 2 Report
- In author presentation section, show clearly the department of the authors.
- In part "2.2. Model specification - Data envelopment analysis", in the second sentence of paragraph three, for the "closed interval", I suggest the symbols "[]" instead of "()".
- In part "3.3. Model Validation—Internal Validity" I would like a clear explanation for the choice of models M5 and M6 and the chosen specifications of inputs and outputs.
Author Response
Dear reviewer,
The suggested improvements have been made into the manuscript.
- In author presentation section, the department of the authors' were clarified
- In part "2.2. Model specification - Data envelopment analysis", the symbol "[]" was used instead of "()" in the second sentence of paragraph three.
- In part "3.3. Model Validation - Internal Validity", the selection of models M5 and M6 was analyzed together with their contribution to the test of internal validity. Also, the specifications for the inputs and outputs excluded from the models were presented.
Thank you for your time and critical interventions, which have been necessary for our work.
Best Regards,
Anastasios Trakakis
Reviewer 3 Report
The paper describes an interesting application of DEA. It is a pity that the work lacks charts showing in a clearer way individual DMUs in the context of CRS and VRS.
- The paper contains a lot of borrowings from [28]: "Total productivity change of Health Centers in Greece in 2016–2018: a Malmquist index data envelopment analysis application for the primary health system of Greece". These citations have not been properly marked in the text of the work, and in many places copying the text has been done carelessly.
- This applies specially to the citation of the DEA model from Coella's work [27], in which quotation marks were opened on page 3 and not closed as in [28], from which a fragment of text was copied.
- Tables 5 and 6 do not describe the X symbol that indicates the presence of a given input in the model. These tables use X in a completely different sense than it is described in the text and in the work [28].
- On page 5 in the third line from the bottom refers to the program DEAP ver2.1 At this point, in the opinion of the reviewer, the last sentence of the summary from page 1, from line 4, should be moved. In addition, reference should be made to work[27].
Author Response
Dear reviewer,
The suggested improvements have been made into the manuscript.
- The paper uses the same data as the paper "Total productivity change of Health Centers in Greece in 2016-2018: a Malmquist index data envelopment analysis application for the primary health system of Greece", but focuses on human resources and their contribution to health center efficiency. The paper was subjected to a plagiarism check by the journal's services before submission, with the following result: "Not many issues were found; most of the % match is due to common phrases/fixed terminology and the bibliography". This was done to prevent unmarking or careless copying of the text of a paper. The reference to the paper "Total productivity change of Health Centers in Greece in 2016-2018: a Malmquist index data envelopment analysis application for the primary health system of Greece" was added in the introduction and some parts of this paper, in which indeed it was missing.
- The quotation mark at the citation of the DEA model from Coelli's work was added.
- "X" symbol in Tables 5 and 6 has been described and explained as it was indicating the presence of a particular input in the model.
- The last sentence of the summary of page 1, line 4, was used on page 5, third line from the bottom, where reference is made to the DEAP ver2.1 program and the bootstrap technique.
- In our opinion, one of the strengths of this study is that it includes the majority of health centers in Greece (155 health centers). Unfortunately, the large number of health centers limits the authors in producing charts and figures that can be presented clearly and legibly.
Thank you for your time and critical interventions, which have been necessary for our work.
Best Regards,
Anastasios Trakakis